



# Insights into Protein Dynamics from $^{15}$N-$^{1}$H HSQC

Erik R.P. Zuiderweg

Department of Biological Chemistry, The University of Michigan, Ann Arbor, MI48109, USA
and
Institute for Molecules and Materials, Faculty of Science, Radboud University Nijmegen, 6525 XZ,

The Netherlands.

*Correspondence to*: Erik R.P. Zuiderweg (zuiderwe@umich.edu)

**Abstract.** Protein dynamic information is customarily extracted from $^{15}$N NMR spin-relaxation experiments. These experiments can only be applied to (small) proteins that can be dissolved to high concentrations. However, most proteins of
interest to the biochemical and biomedical community are large and relatively insoluble. These proteins often have functional conformational changes, and it is particularly regretful that these processes cannot be supplemented by dynamical information from NMR. We ask here whether (some) dynamic information can be obtained from the $^{1}$H line widths in $^{15}$N-$^{1}$H HSQC spectra. Such spectra are widely available, also for larger proteins. We developed computer programs to predict amide proton line widths from (crystal) structures. We aim to answer the following basic questions: is the $^{1}$H linewidth of a HSQC cross
peak smaller than average because its $^{1}$H nucleus has few dipolar neighbors, or because the resonance is motionally narrowed? Is a broad line broad because of conformational exchange, or because the $^{1}$H nucleus resides in a dense proton environment? We calibrate our programs by comparing computational and experimental results for GB1 (58 residues). We deduce that GB1 has low average $^{1}$HN order parameters (0.8), in broad agreement with what was found by others from $^{15}$N relaxation experiments (Idiyatullin et al., 2003). We apply the program to the BPTI crystal structure and compare the results with a $^{15}$N-
$^{1}$H HSQC spectrum of BPTI (56 residues) and identify a cluster of conformationally broadened $^{1}$HN resonances that belong to an area, for which millisecond dynamics has been previously reported from $^{15}$N relaxation data (Szyperski et al., J. Biomol. NMR 3, 151-164, 1993). We feel that our computational approach is useful to glean insights into the dynamical properties of larger biomolecules for which high-quality $^{15}$N relaxation data cannot be recorded.



## 1. Introduction

Providing evidence that proteins are dynamical rather than rigid molecules is a major contribution of solution NMR to structural biology and molecular biophysics (Kay, 1998). The dynamic information is extracted from NMR spin-relaxation experiments, mostly of the amide nitrogen (Kay et al., 1989), but also from methyls (Nicholson et al., 1992) (Lee et al., 2000) and from carbonyl (Wang et al., 2006). The amide nitrogen relaxation experiments are the easiest to implement, require just $^{15}N$ isotopic labeling, are potentially complete, and analysis software is broadly available (Mandel et al., 1995). But these experiments are rather insensitive (especially the $^{1}H \rightarrow {}^{15}N$ NOE) and can therefore only be applied to (small) proteins that can be dissolved to high concentrations.

However, most proteins of interest to the biochemical and biomedical community are large, and cannot be studied with $^{15}N$ dynamics measurements. This is particularly regretful, because larger proteins, more than smaller, often display functional conformational changes which cannot be supplemented by dynamical information from NMR. Moreover, considering the insight that fast dynamics contributes to configurational entropy (Akke et al., 1993) (Yang and Kay, 1996) (Lee et al., 2000), lack of measurement of dynamics also results in the lack of (experimental) understanding of the protein's thermodynamics.

Due to the paucity of dynamical measurements of the proteins of interest to the biochemical and biomedical community, the above "dynamics awareness" has not generally taken hold in that important area of science and medicine, which I find regretful. However, $^{15}N$-$^{1}H$ HSQC and TROSY HSQC experiments can be recorded for (very) large proteins ( $< 300$ kDa) at relatively low concentrations ($< 50$ uM) . This experiment contains conformational dynamics information in the intensity and line widths of its cross peaks. In this contribution, we explore if we can harvest (some) of the dynamical information from that data, without the need for specific "relaxation" experiments and/or labeling strategies (Gardner et al., 1997).

The study of a protein by solution NMR usually starts by recording such a $^{1}HN$-$^{15}N$ HSQC or TROSY HSQC spectrum of the sample. Such a dataset is seen as a "fingerprint" of the protein, from which important molecular parameters can immediately be gleaned. For instance, an experienced NMR spectroscopist recognizes from such a spectrum whether the sample is pure, the protein is (mostly) folded, whether it aggregates, or whether it has multiple conformations. Looking into more detail, the spectroscopists finds that the $^{1}HN$ linewidths of the cross peaks will differ from each other (not when a dominating windowing function was used). The spectroscopist may want to infer that lines narrower than average are due to fast dynamics, and broad lines to conformational exchange processes. But such inferences are not warranted. Amide proton linewidths may be affected by a plethora of mechanisms, which we will try to unravel in this work. Only after that we can answer the question: is a narrow line narrow because it has few dipolar neighbors, or is it motionally narrowed? Is a broad line broad because of conformational exchange, or because it has a dense proton environment?








**Figure 1.** *Section of the 600 MHz $^{15}$N-$^{1}$H HSQC spectrum of GB1 at 3 $^{0}$C, pH 6.5.*

*Processed with 1 Hz EM in $t_2$, $cos^2$ in $t_1$. The assignments were taken from BMRB entries 7280, 25909 and 26716.*

*The numbering is as in PDB entry 6c9o.pdb. The cross sections show the Gaussian line shape fits as carried out*

*by Sparky(Goddard and Kneller, 2000)*

Let us take a look at the $^{15}$N-$^{1}$H HSQC spectrum of GB1 (Figure 1). We take this protein as our "calibration" case. The assignments are available at the Biological Magnetic Resonance Bank, while high-resolution crystal structures are

available in the Protein Data Bank. The intensities of the cross peaks in the spectrum vary by a factor of 3 and the $^{1}$HN linewidths vary by a factor 2 (see Table S1 in the Supplemental Materials). The spectrum shows several doublets (e.g. N8), due to the 3JHNHA coupling. What causes the variation in $^{1}$HN line width? There are several possibilities. First, (unresolved) 3JHNHA contributes to the measured linewidth. Second, the dipolar proton environment of each amide proton varies. This causes differences in relaxation rates, $R_2$ $^{1}$HN-$^{1}$HX. Third, anisotropic rotational diffusion may cause differences in the line

widths. Fourth, the amide proton resonances could be life-time broadened by mass exchange with water. Fitfh, last but not least, local dynamics could affect the line widths, either by narrowing (fast local dynamics) or broadening (conformational exchange dynamics on the ms –ms timescale), e.g. di-sulfide isomerization, or general conformational flexibility.

We will address these points one by one. Can anisotropic rotational diffusion cause the line width differences? GB1 is an ellipsoid with an long/short axis ratio of 1.8 (Idiyatullin et al., 2003). We calculate from the classical Woessner equations

(Woessner, 1962) that $R_2$ for the $^{1}$HN-$^{15}$N dipolar interaction varies +/- 12 % when considering different angles from a relaxation vector to the diffusion axes. But that is for individual relaxation vectors – the $^{1}$HN-$^{1}$HX relaxation vectors contributing to the dipolar $R_2$ relaxation of a particular $^{1}$HN point in different directions; so, in practice, the small orientational effects will mostly cancel.

The intrinsic (unprotected) amide proton exchange rate is given by the empirical relation (Englander et al., 1972) :

$$k_{ex} = \frac{\ln 2}{200}\left[10^{pH-3} + 10^{3-pH}\right] \times 10^{0.05T} \qquad [1]$$

where $T$ is in $^{0}$C and $k_{ex}$ in min$^{-1}$.

From the experimental parameters of the spectrum (3 ºC, pH 6.5) we calculate from Eq [1] a 0.26 s$^{-1}$ exchange rate, giving rise to a 0.1 Hz life-time broadening for unprotected amide proton resonances. Amide protons engaged in H-bonds within the

protein will exchange much slower, with even less broadening. We find that variation in amide proton exchange is not significant for this spectrum.



In principle, the specific proton environment of each amide proton is known from the (high resolution) structure of GB1. Hence, the dipolar $R_2$ relaxation of $^1$HN due to its surrounding protons can be calculated, given the three-dimensional structure.

The (unresolved) $^3J_{HN-H\alpha}$ scalar coupling  is also knowable and can be obtained from the structure as well,  using the following Karplus equation (Lee et al., 2015)

$$^3J_{HN-H\alpha} = 8.83\cos^2\left(\theta - 60\right) - 1.29\cos\left(\theta - 60\right) + 0.20 \qquad [2]$$

where $\theta$ is the dihedral angle spanned by C'-N-Ca-C'.


Summarizing, we have a handle on variables 1 through 4 that affect the $^1$HN line width. Hence, making a calculation of these variables, and comparing the resulting calculated linewidth with the experimental (reduced, see below) linewidth, should uncover the presence (or not) of the dynamic properties of the protein, in a sequence-specific fashion.






## 2. Theory of $R_2$ $^1$HN-$^1$HX relaxation

Measuring $R_2$ relaxation in proteins has been taken on by Bodenhausen and co-workers (Boulat and Bodenhausen, 1993) (Segawa and Bodenhausen, 2013). Their work has been focusing on obtaining "pure" $R_2$ rates for resolved $^1$H resonances in
1D NMR spectra, using extremely selective pulses and selective spinlocks. As far as we know, they have not published a linewidth fit of a complete HSQC spectrum, as we are trying to do here.

As virtually all $^1$H resonances are resolved in the small proteins GB1 and BPTI, the pure $^1$HN-$^1$HX $R_2$ relaxation rate as measured from the cross peaks in a $^{15}$N-$^1$H HSQC is given by the $R_2$ relaxation rate for un-like spins (Goldman, 1988)

$$R_2^{H(H)} = \frac{1}{20}\left(\frac{\mu_0}{4\pi}\frac{\gamma_H\gamma_H\hbar}{r_{HH}^3}\right)^2 \times \left\{5\tau_c + \frac{9\tau_c}{1+\omega_H^2\tau_c^2} + \frac{6\tau_c}{1+\left(2\omega_H\right)^2\tau_c^2}\right\} \qquad [3]$$

where $\mu_0$ is the permittivity of space, $\gamma$ are the gyromagnetic ratios, $\hbar$ Planck's constant divided by $2\pi$, $\omega$ the resonance frequency, and $\tau_c$ the rotational correlation time.

Exceptions to equation [3] will arise when the interacting protons have identical chemical shifts (within linewidth). So this may happen for a few $^1$HN-$^1$HN dipolar pairs. In that case one should use the "identical" equation (Goldman, 1988).

$$R_2^{H(H)} = \frac{1}{20}\left(\frac{\mu_0}{4\pi}\frac{\gamma_H\gamma_H\hbar}{r_{HH}^3}\right)^2 \times \left\{9\tau_c + \frac{15\tau_c}{1+\omega_H^2\tau_c^2} + \frac{6\tau_c}{1+\left(2\omega_H\right)^2\tau_c^2}\right\} \qquad [4]$$

An equation describing a smooth transition between identical and non-identical spins has been given by (Goldman, 1988).

However, it is often difficult to measure a "pure" $R_2$ for HN, because it is affected by anti-phase relaxation due to the
scalar-coupled $^1$HA (Peng and Wagner, 1992):

$$\sigma^{HN}(t) = H_x^{HN}\cos\left(\pi J_{HNHA}t\right)\exp\left(-R_2^{HN}t\right) + 2H_y^{HN}H_z^{HA}\sin\left(\pi J_{HNHA}t\right)\exp\left(-\left(R_2^{HN}+R_1^{HA}\right)t\right) \qquad [5]$$

and therefore, one measures

$$R_2^{HN-eff} = f_{IP}R_2^{HN} + (1-f_{IP})*(R_2^{HN}+R_1^{HA}) \qquad [6]$$

where $f_{IP}$ is the fraction of time when the coherence is in-phase. In the case of the GB1 HSQC spectrum, where we used a 227 ms $t_2$ acquisition period, we calculate that for $3J_{HNHA} > 3$ Hz, $\left(1-f_{IP}\right)$ is close to 0.5. Hence the $R_1$ rate of the $^1$HA does





come into play up to 50% and affects the $^1$HN linewidth. The key question is now whether the HA $R_1$ rate is the "selective" or "unselective" $R_1$. For macromolecules, the difference between the rates is very large (~10 s$^{-1}$ for selective, ~0.2 s$^{-1}$ for unselective).

The literature is unanimous to state that it is the fast "selective" rate. In the case of (Boulat and Bodenhausen, 1993)'s careful $^1$H R$_2$ measurements, this is certainly the case: they employ extremely selective pulses that excite a single $^1$H resonance in the NMR spectrum. The scalar coupling, if allowed to develop, then affects the z-magnetization of just one other proton.

All other $^1$H spins are unperturbed as serve as a cross-relaxation bath for that latter proton, driven by the large spectral density term $J(0)$ (actually $J\left(\left|\omega_{HN} - \omega_{HA}\right|\right)$ ) (Goldman, 1988).:

$$R_1^{HH-SEL} = \frac{1}{20}\left(\frac{\mu_0}{4\pi}\frac{\gamma_H\gamma_H\hbar}{r_{HH}^3}\right)^2 \times \left\{2\tau_c + \frac{6\tau_c}{1+\omega_H^2\tau_c^2} + \frac{12}{1+\left(2\omega_H\right)^2\tau_c^2}\right\} \qquad [7]$$

But our case is different. For the $^1$HN linewidth in a HSQC, we need to consider the magnetic environment of the scalar coupled $^1$HA during the FID. In fact, all $^1$H magnetizations are in the $xy$ plane, and thus saturated, by the last $^1$H 90 degree pulse. Hence the relaxation bath is "hot" and cross relaxation between the $^1$HA and other protein protons is expected to be slow. We should thus expect that the unselective, slow, $^1$HA $R_1$ rate is at play in this case (Goldman, 1988).:

$$R_1^{HH-UNSEL} = \frac{1}{20}\left(\frac{\mu_0}{4\pi}\frac{\gamma_H\gamma_H\hbar}{r_{HH}^3}\right)^2 \times \left\{\frac{3\tau_c}{1+\omega_H^2\tau_c^2} + \frac{24}{1+\left(2\omega_H\right)^2\tau_c^2}\right\} \qquad [8]$$

This argument may need refinement. After all, the fast-HSQC experiment we employed, was designed to return the water magnetization to +z during the FID. Therefore, most of the $^1$HA magnetizations will also be in +z and their $R_1$ rate will be affected by other aliphatic protons that are saturated. All in all, the situation can become dependent on the detail of the experiments and chemical shift distributions. In the Results section we will employ a "see what fits best" approach.

The entire in-phase / anti-phase and ensuing "selective" / unselective issues is of course moot when one studies perdeuterated proteins. But there are other methods to avoid the problem. Methods to measure pure R$_2$ in the presence scalar couplings have been developed by Bodenhausen and co-workers (Boulat and Bodenhausen, 1993) (Segawa and Bodenhausen, 2013), and by Morris and co-workers (Aguilar et al., 2012). In the original Bodenhausen approach, one obtains a pure HN R$_2$ when selectively exciting a single amide proton resonance, and selectively spin locking it with a very weak r.f. field.

The relaxation rate of that spin-locked resonance is unambiguously described by (Brüschweiler, 1991):

$$\frac{1}{T_{1\rho}} = \frac{1}{20}\left(\frac{\mu_0}{4\pi}\frac{\gamma_H\gamma_H\hbar}{r_{HH}^3}\right)^2 \times \left\{\frac{5\tau_c}{1+\omega_{rf-eff}^2\tau_c^2} + \frac{9\tau_c}{1+\omega_H^2\tau_c^2} + \frac{6\tau_c}{1+\left(2\omega_H\right)^2\tau_c^2}\right\} \qquad [9]$$

Here, $\omega_{rf-eff} = \sqrt{\omega_{rf}^2 + \Delta\omega^2}$ , where $\Delta\omega$ is the offfeset between the r.f. carrier and the resonance frequency.

Offset issues also lead to (Massi et al., 2004)

$$R_{1\rho}^{HN-eff} = R_{1\rho}\cos^2\theta + R_1\sin^2\theta \qquad [10]$$

where $\theta = arctg\left(\frac{\Delta\omega}{\omega_{rf}}\right)$. (In the Bodenhausen expriment, the r.f. carrier is on the locked resonance, and off-resonance

effects do not come into play).

However, this experiment is rather impractical for proteins; there is not enough amide proton resolution in the spectrum of even a small protein such as GB1 to select more than a few resonances individually. We are therefore using a variation of this approach. As suggested by Dr. G. Morris (Manchester), we selectively excite all amides, and spin lock them

at high power. This avoids the issues with the scalar coupling. Also, because $\omega_{rf-eff}^2\tau_c^2 << 0$ , equation [9] becomes identical

to equation [3]. Regretfully, it brings in a new issue: are the spin-locked protons relaxing according to the "like" or "unlike" equations? Common wisdom is that since the aliphatic protons are not-spin-locked, amides and aliphatics are "unlike" and follow equation [9]. We had some doubt and elected to test this experimentally. We compared the apparent $R_{1rho}$ rate for the amide of F52 (9.5 ppm), between two spin-lock fields at 5 kHz and 500 Hz. According to the structure, the dipolar relaxation

of this amide is dominated by the HA of residues 52 and 51. Hence the 5 kHz r.f. field "hits" those HA whereas the 500 Hz r.f. field does not. And yet, the $T_{1rho}$ is identical for both locking fields (see Figure 2) with the same factor of 2.0 drop in intensity between 50 and 75 ms of spinlocking (serendipitous). This convinced us that the excitation profile, and not the locking field strength., determines what are "like" or "unlike" spins.

**MAGNETIC RESONANCE**
Open Access Discussions

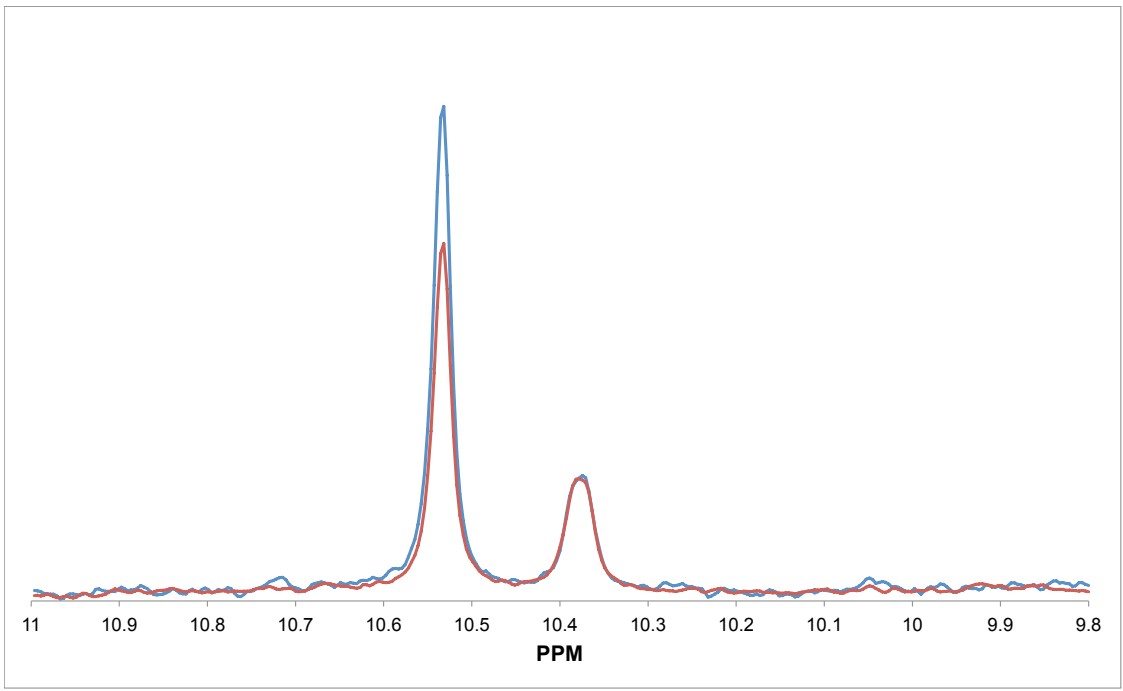


*Figure 2A*. *The peak intensity of Trp43-HE1 (10.53 ppm) and Phe52-HN 10.39 ppm after a 50 ms spinlock (red) and after a 75 ms spinlock (blue), of 5270 Hz, at 10.39 ppm. The red trace was multiplied by 0.5. The $T_{1rho}$-HSQC experiment (see supplemental data) was used in 1D mode. The FIDs were multiplied by 5 Hz LB.*



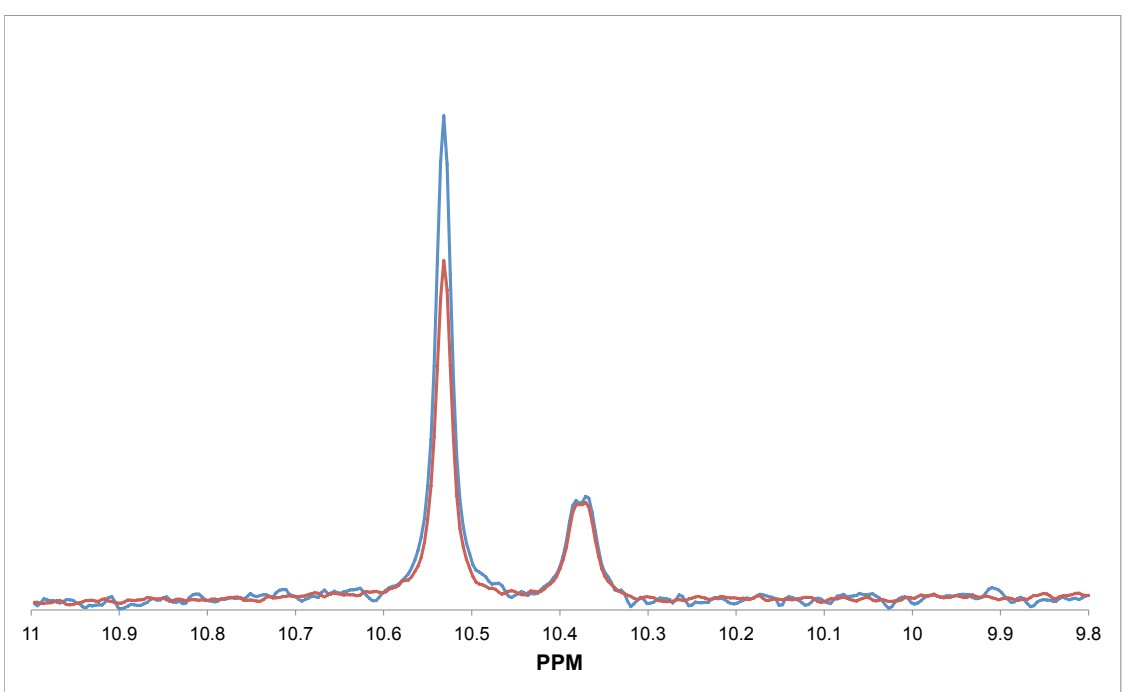

*Figure 2B. The peak intensity Trp43-HE1 (10.53 ppm) and Phe52-HN (10.39) ppm after a 50 ms spinlock (red) and a 75 ms spinlock (blue) of 527 Hz, at 10.39 ppm. The red trace was multiplied by 0.5. The $T_{1rho}$-HSQC experiment (see supplemental data) was used in 1D mode. The FIDs were multiplied by 5 Hz LB.*

For those amide protons, for which dipolar partners such as other amide protons, aromatic ring protons, Gln, Asn and Arg sidechain protons are also selected and properly spin-locked, the relaxation will be given by a "like" spin equation (Bothner-By et al., 1984)

$$\frac{1}{T_{1\rho}} = \frac{1}{20}\left(\frac{\mu_0}{4\pi}\frac{\gamma_H\gamma_H\hbar}{r_{HH}^3}\right)^2 \times \left\{\frac{9\tau_c}{1+\omega_{rf-eff}^2\tau_c^2} + \frac{15\tau_c}{1+\omega_H^2\tau_c^2} + \frac{6\tau_c}{1+\left(2\omega_H\right)^2\tau_c^2}\right\} \qquad [11]$$

With $\omega_{rf-eff}^2\tau_c^2 << 0$ this reduces to the "like" spin relaxation rate [5].



Hence, one can predict with great certainty which equation should be used for which spin-pair. Furthermore, the spin-lock virtually eliminates broadening by conformational exchange, unless $\omega_{rf-eff}$ being in the kHz range, becomes in "resonance" with conformational/chemical exchange processes at the same timescale.

All in all, the theory behind the T$_{1rho}$ experiment is more robust than the theory behind the cross peak linewidth. Therefore, we will use the T$_{1rho}$ experiment and calculations as our departure point for further calculations.

Even in small proteins, many protons interact magnetically. In our programs, described in the appendix, we find typically that 40 protons are present in a 6 Å sphere around an amide proton. All R$_2$ or R$_{1rho}$ relaxation rates of these $N$ other protons $j$ for an amide proton $i$ will co-add if the relaxation vectors $ij$ diffuse independently from each other:

$$R_2^{i-total} = \sum_{j \neq i}^{j=N} R_2^{ij} \qquad [12]$$

Taking a larger sphere of interacting protons does not significantly change the summation (see Table S3 in the Supplemental materials)

Obviously, the assumption underlying equation [12] cannot be correct, because the interacting protons in a protein are *not* diffusing independently. One has to consider dipole-dipole cross-correlated $R_2$ relaxation (also called relaxation interference). However, we can show that relaxation interference is almost completely canceled in multi-spin systems, and can be neglected as a source for large deviations of equation [12] (see Appendix).





## 3.Results and Discussion

Our experimental / computational approach is the following. First we analyze the $T_{1rho}$ experiments for GB1, and extract the effective rotational correlation time;  then we use that correlation time to analyze the GB1 amide proton linewidth
and decide, experimentally, whether the $R_1$ relaxation rate for $^1HA$ contributing to the effective $R_2$ relaxation rate of $^1HN$ in the HSQC, Equation [6] is (closer) to the "selective" or "unselective" $R_1$.

Subsequently, we use what we have learned from GB1 to calculate the HSQC linewidths for BPTI, and analyze the results for dynamical content, and compare with the literature.

### 3.1 Calibration: GB1 $T_{1rho}$

For GB1, we collected not only the HSQC spectrum of Figure 1, but also a series of spectra, in which we aim to measure the $^1HN$ $R_{1rho}$ rate by using an amide-selective excitation pulse followed by a fairly strong spinlock field ($\omega_{rf}$ = 5.3 KHz ) of varying durations, followed by a HSQC read-out. In this experiment, called semi-selective $T_{1rho}$-HSQC, the $^3J_{HNHa}$ scalar coupling is suppressed. The pulse sequence and the relaxation data obtained are shown in the supplemental materials.
With few exceptions, the relaxation data can be fitted with a single exponential with a $R^2 > 0.95$.

For the computations, there are several high-resolution crystal structures 6c9o (V29SeM; 1.2 Å resolution), 6che (A34Sem; 1.1 Å ), 6cne (V29SeM; 1.2 Å ) and 6cpz (I6Sem; 1.12 Å). SeM is seleno-methionine.  Inspection of the structures suggests that the mutation at I6 is the least intrusive on the structure. This mutant is a dimer in the crystal. We use only chain "A" for our calculations. The results for chain "B" are not significantly different. The proton coordinates were added by the
routine Molprobity (Williams et al., 2018).

The effective rotational correlation time for the GB1, which is a prolate ellipsoid, was determined to be 6.5 ns from an extensive analysis of $^{15}N$ relaxation data acquired at 5 $^0C$ (Idiyatullin et al., 2003). Using the empirical relation from (Daragan and Mayo, 1997) we extrapolate from the experimental data at 5 $^0C$ that  $\tau_c$ = 7  ns at 3 $^0C$. When using the same empirical equation directly, we find for GB1 that $\tau_c$ = 8.9  ns at 3 $^0C$ (see also Appendix).

At the outset, we note that by fitting the experimental and calculated $T_{1rho}$ data, we can only determine an effective rotational correlation time, which is the product $\left\langle \overline{S_{HN}^2} \tau_c \right\rangle$ where the brackets indicate average over residues, and $\overline{S_{HN}^2}$ is an average order parameter for each $^1HN$ describing the motions of $^1HN$-$^1HX$ relaxation vectors (in terms of both  distance and angular fluctuations).

As a start, we used the 7 ns effective correlation time to compute $R_{1rho}$ for GB1. The $^1HN$ $R_{1rho}$ rates due to 6 Å sphere
of protons around it, were calculated from equation  [9] or [11], depending on whether the other spin was spin locked or not. Because  $\omega_{rf-eff}^2 \tau_c^2 << 0$ we needed not to concern ourselfs with offset effects (see Eq [10]).



The comparison between experimental and calculated $R_{1rho}$ rates is shown in Figure 3. One sees that the range and median of the computed and experimental $R_{1rho}$ correspond very well, indicating that we have chosen the correct effective rotational

correlation time for the calculations. Actually, we optimized the effective correlation time, by minimizing the RMSD between measured and calculated $R_{1rho}$. At this point, it is important to recall that we can only optimize the effective correlation time; it may be well the theoretical 8.9 ns multiplied by an $\overline{S^2_{HN}}$ of 0.79. Indeed, one could do more experiments e.g. the analysis of $^{15}$N relaxation, to get a better handle on that correlation time, but, as explained in the introduction, we would like to find a way to obtain dynamical information without such experiments. While the range of calculated and experimental $R_{1rho}$ rates

correspond well, the actual correlation is poor. For calculated data points larger than the experimental ones can be explained by  low order parameters. Experimental data points larger than computed ones are harder to explain. In $T_{1rho}$ the latter cannot be, in general, caused exchange broadening, as it is suppressed by the spin lock.

For now we chose not to be concerned by these issues as we want to use the experiment to help us decide the (effective) rotational correlation time.

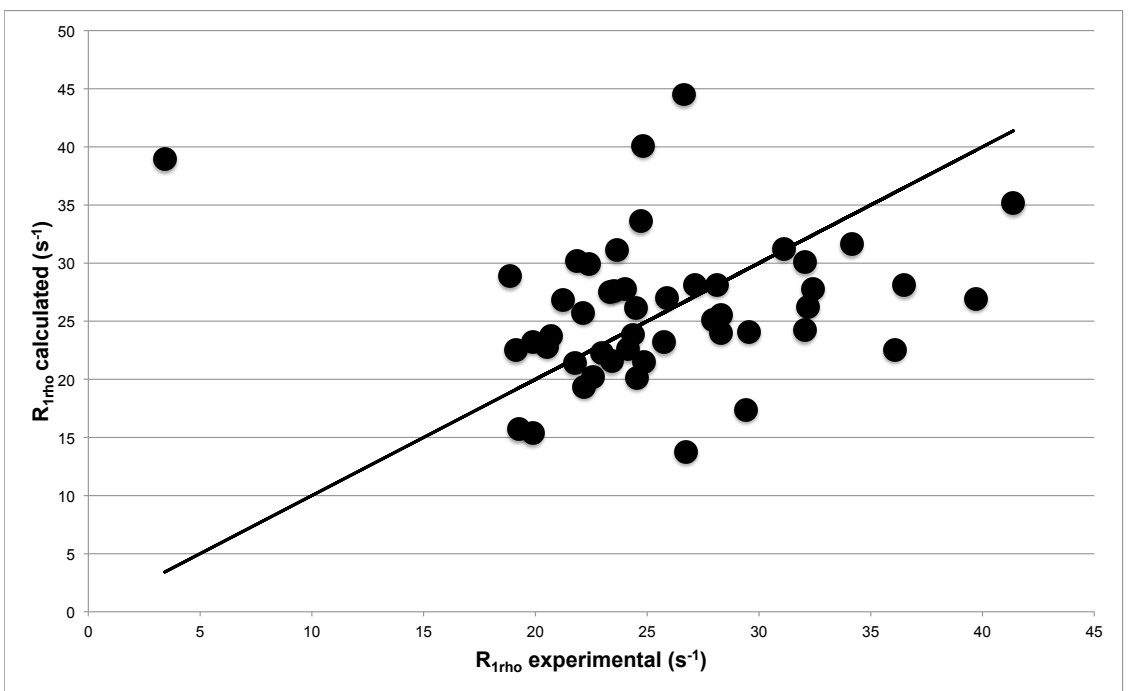


*Figure 3. Experimental (600 MHz) and calculated $R_{1rho}$ rates for the $^1HN$ resonances of GB1, at 3 $^0C$. The line y=x is shown. The calculations were based on structure 6cpz, using an effective correlation time of 7 ns.*



## 3.2 Calibration: GB1 HSQC

255        With the $R_{1rho}$ data analyzed and the effective correlation time estimated, we turn our attention to the HSQC spectrum itself.  The HSQC spectrum in Figure 1, processed with 1 Hz exponential window in $t_2$, shows many of the $^3J_{HNHA}$ scalar couplings, which closely correspond to the scalar couplings we compute from the crystal structure using the Karplus equation [2]. The Sparky software (Goddard and Kneller, 2000) does not fit resolved doublets as a pair, but as a single resonance, as shown in the cross sections shown in Figure 1. The fits, using a Gaussian lineshape, were individually inspected and found to

be excellent, with an estimated uncertainty of less than a 1 Hz.

        Before we can make comparisons between experimental $R_2$ and computed $R_2$ data, we have to correct the measured $^1HN$ line widths for several effects. First, we subtract the computed scalar couplings. Second, we need to consider $^1HN$ dipolar relaxation due to the amide nitrogen, chemical shift anisotropy relaxation and field inhomogeneity. We calculate that for $\tau_c$ =7 ns, the $^1HN$ dipolar interaction with $^{15}N$ accounts for ~2 Hz, that the $^1H$ CSA contributes ~ 0.2 Hz at 600 MHz (using CSA

values from (Loth et al., 2005)), while field inhomogeneity typically is limited to 1 Hz. We decoupled $^{13}CO$ during data acquisition, but the $^2J_{HNCA}$ of  2 Hz (Schmidt et al., 2011) should also contribute to the $^1HN$ linewidth. We thus are inclined to subtract 5 Hz from the apparent experimental $^1HN$ line widths in addition to the 1 Hz due to the window function (total 6 Hz). What is left is what we call the "reduced experimental line width", which *should* consist of just the sum of the $^1HN$-$^1HX$ dipolar line widths, affected by antiphase relaxation due to the $^3J_{HNHA}$ and potentially affected by the fast and/or  slow dynamics we

try to uncover. However, we find, by simulation, that unresolved scalar couplings add a full 1 Hz less to the linewidth than expected.  When assuming that the instrument was well-shimmed, it is thus reasonable to estimate that one should subtract just 2-3 Hz from the observed linewidths. We will determine what is best from the calculations.

        For Figure 4  we used Equations 3, 6 7 and 8, with the effective  $\tau_c$ =7 ns obtained by the $T_{1rho}$ experiments taking

into account all protons in a sphere of 6 Å around the individual amide protons. We compute the selective and unselective $R_{1HA}$ also from all $^1HA$-$^1HX$ interactions within 6 Å in the crystal structure, co-adding all relaxation rates [7], not taking into account cross correlated relaxation on basis of the same arguments outlined for $R_2$ (see Appendix).   The fractions in-phase for Equation [6] were calculated from the calculated scalar couplings and the acquisition time (227 ms), and were found to vary between 0.43 and 0.65. Hence taking the antiphase relaxation into account is necessary.


**MAGNETIC RESONANCE**
Open Access Discussions

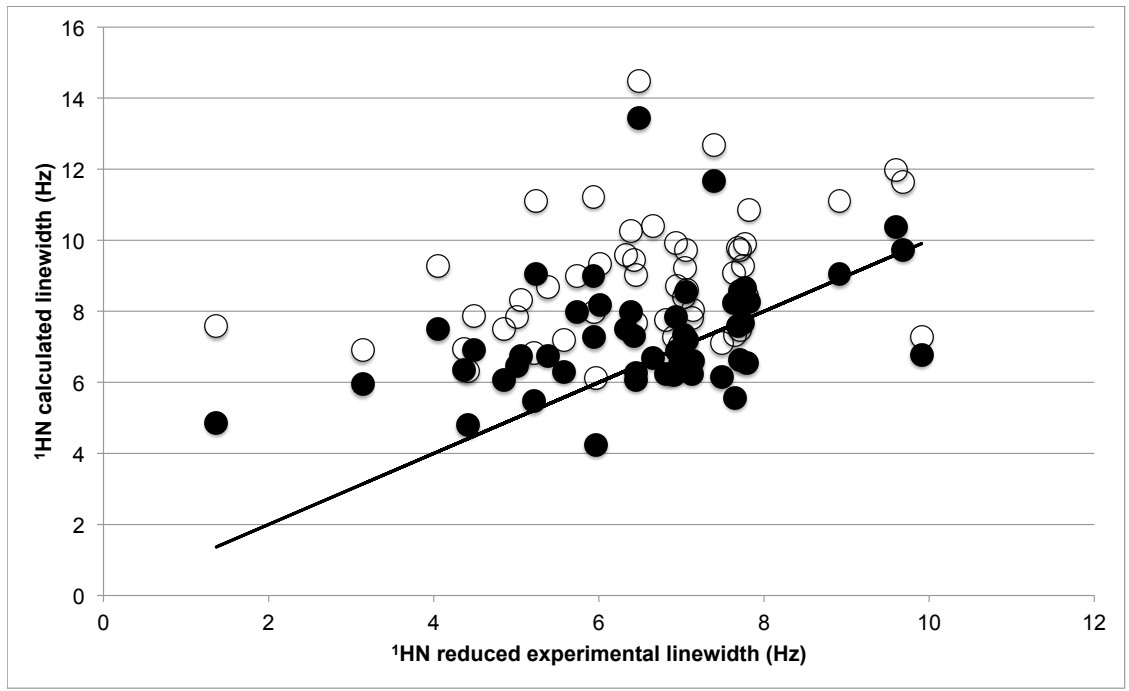

> ***Figure 4A***. *Calculated $^1$HN linewidths for GB1, using Equations 3 and 6,* $\tau_c$ *=7 ns*
>
> *The experimental linewidth was reduced by the scalar couplings and 3Hz (see text).*
> *For the filled circles, we used the "unselective" $^1$HA $R_1$ (Eq 8). The RMSD with the experimental values is 1.91Hz. For the open circles we used the "selective" $^1$HA $R_1$ (Eq 7).*
> *The RMSD with the experimental values is 3.0 Hz.*


In Figure 4A we show the results of these calculations. For panel A, we subtracted 3 Hz from the experimental linewidth (in addition to $^3$J$_{HNHA}$) as outlined above. It is clear that, when doing that, the linewidths calculated taking the unselective HA $R_1$ into account (filled circles), are on average too large (RMSD 1.91Hz). It is also clear that the calculations using the selective HA $R_1$ into account turn out worse ( open circles; RMSD 3.00). For Figure 4B , we subtracted just 2 Hz

from the experimental linewidth (in addition to $^3$J$_{HNHA}$). Now the median value of the "unselective" calculation corresponds better to that of the reduced experimental values and we obtain a RMSD of 1.75 Hz. For the "selective" calculation we obtain RMSD 2.29 Hz.

This is what we set out to resolve. The theory (and my consultancies with several colleagues) does not establish unambiguously which HA $R_1$ rate is to be used; but the comparison between experiment and calculation does. Clearly, we



need the "unselective" HA  $R_1$ in this experiment. In practice, this rate is so small, that the anti-phase relaxation is within a

few tenths of s$^{-1}$ within the in-phase rate. Hence, we do not need to worry about in-phase / anti-phase issues. The situation will

be different when using a HSQC pulse sequence using selective amide proton pulses throughout , not exciting anything else

(Gal et al., 2007) . In that case, the scalar-coupling-induced HA z-magnetization perturbation during the FID would be in an

unperturbed aliphatic spin bath, and the fast "selective" HA $R_1$ would be in effect.

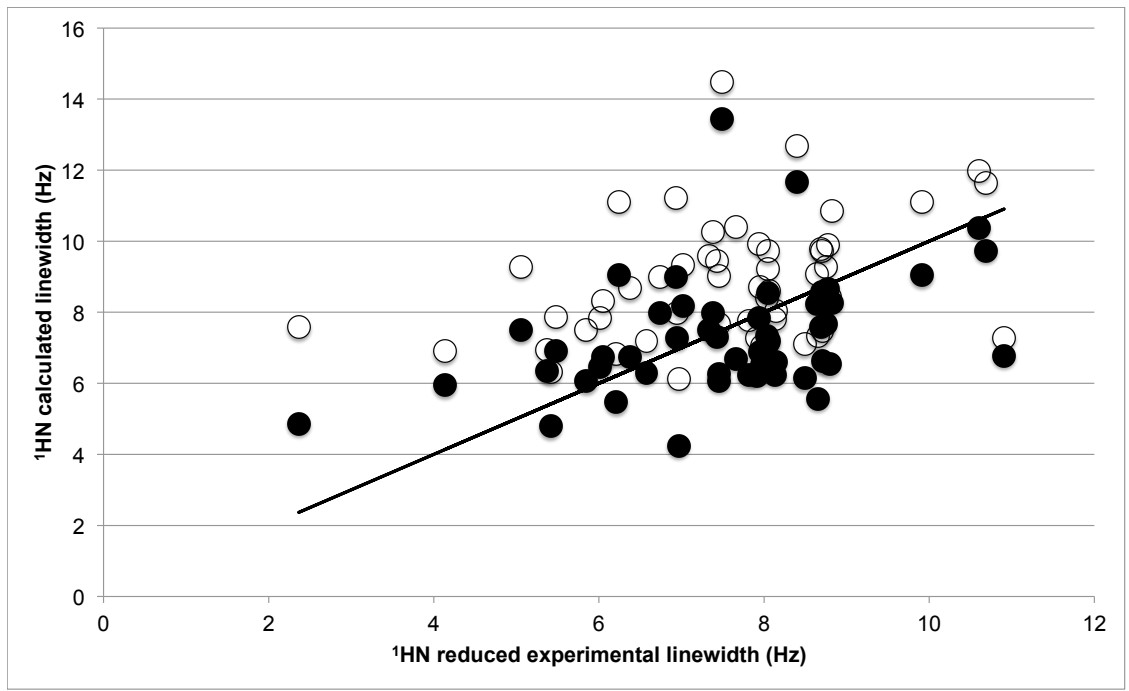


> ***Figure 4B***. *Calculated $^1$HN linewidths for GB1, using Equations 3 and 6, $\tau_c$ =7 ns*
>
> *The experimental linewidth was reduced by the scalar couplings and 2 Hz (see text).*
>
> *For the filled circles, we used the "unselective" $^1$HA $R_1$ (Eq 8). The RMSD with the experimental values is 1.75Hz. For the open circles we used the "selective" $^1$HA $R_1$ (Eq 7) .*
>
> *The RMSD with the experimental values is 2.29 Hz.*


As explained before, we cannot determine a real correlation time from this fitting procedure. Rather we determine the product

$\left\langle \overline{S_{HN}^2} \tau_c \right\rangle$ to be 7 ns. If we take $\tau_c = 8.9$ ns at 3 $^0$C as calculated from the empirical relation (Daragan and Mayo, 1997) for





$\tau_c$ , we would obtain that $\overline{S^2_{HN}}$ equals 0.79. Parenthetically, we note that we have independently carried out an analysis of

protein rotational correlation times as available in the literature, and found those to closely follow the empirical relation (Daragan and Mayo, 1997) (see appendix).

How does the estimated $\overline{S^2_{HN}}$ compare with literature values? Obviously, there are no such values determined, but there is an comprehensive paper of (Idiyatullin et al., 2003) calculating [15]N order parameters using several different approaches. Using the extended Modelfree method (Clore et al., 1990), they obtain an average order parameter of 0.70, while using their

own method, they obtain an average of 0.62.  From this it is suggested that GB1 is a rather dynamical molecule, and one may argue that one sees that back in the [1]HN relaxation as well.  However, we note that (Idiyatullin et al., 2003) obtain a rotational correlation time $\tau_c$ for the GB1 prolate ellipsoid of 6.5 ns from the [15]N relaxation data acquired at 5 [0]C,  while the empirical relation from the same lab would predict $\tau_c$ 8.6 ns. Even so, both methods point to a quite dynamic GB1 protein, even at temperatures as low as 3 0C.

In summary, for GB1, we have established by comparing $T_{1rho}$ and HSQC linewidth data, that one can compute a reasonable range of [1]HN $R_2$ rates using just a single equation for unlike spins. When assuming a rotational correlation time as predicted by the literature, we *predict*  from comparing the calculated and experimental [1]HN linewidths that the average [1]HN-[1]HA order parameter must be 0.79, indicating much internal motion.  Others have come to similar conclusions analyzing [15]N relaxation data (Idiyatullin et al., 2003). With this, we have arrived at our goal: we show that we *can* extract motional

information from the [1]HN linewidths in a HSQC spectrum by making simple calculations based on a crystal structure. In the case of GB1 this is overall motional narrowing. In BPTI, as we will see below, we can also extract conformational exchange line broadening that is not immediately apparent from the spectrum itself.






## 3.3 Application: BPTI

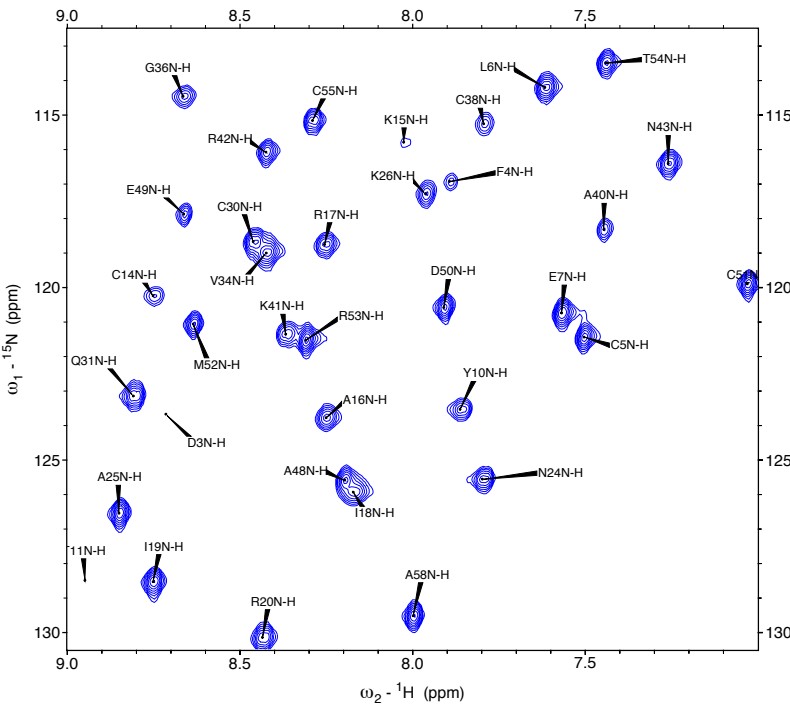

**Figure 5.** *Section of a 500 MHz $^{15}$N-$^1$H HSQC for BPTI, 30 $^o$C, pH 5.8 , processed with a 1Hz exponential window in $t_1$ and $t_2$.*


Assignments and the time-domain data for a $^{15}$N-$^1$H HSQC spectrum of bovine trypsin inhibitor (BPTI) recorded at 30 $^o$C, pH 5.8  are available at the Biological Magnetic Resonance Bank, while a 1.3 Å resolution crystal structure (9PTI.PDB) is available in the Protein Data Bank. We processed the time-domain data with a 1 Hz exponential window in $t_2$. At the contour

level of Figure 5, several peaks are missing, indicating intensity dispersion. Table S2 shows a 30-fold range for S/N and an 8-fold range for the linewidth, much more than for GB1. Significantly, and in contrast to GB1 at 3 $^0$C, this processed spectrum does *not* show any resolved $^3$J$_{HNHA}$ couplings at 30 $^0$C. Nevertheless, the protein is not perdeuterated. According to three sources, the correlation time should be between 2.5 and 3.5 ns, much less than the 7 ns correlation time of GB1 at 3 $^0$C. (Beeser et al., 1997),(Daragan and Mayo, 1997) and  (Sareth et al., 2000). Why are these doublets missing? From Equation [1]we

calculate a 1.15 s$^{-1}$ mass exchange rate, giving rise to maximum   broadening of ~ 0.4 Hz for unprotected amide proton resonances, so that cannot be the reason. We can, a priori, already assume that the sample must have been aggregated. Indeed, MRD studies suggest that, at high concentrations, BPTI can form  decamers in solution (Gottschalk et al., 2003).  As it turns





out, this sample of BPTI mimics a much larger molecule, and provides an excellent opportunity to test our method on a "large" protein.

350        In Figure 6, we show the comparison of experimental and calculated [1]HN linewidths for BPTI. We subtracted, besides the calculated scalar couplings, 2 Hz from the linewidths as reported by Sparky ([15]N-H dipolar, [1]HN CSA and shimming). For the calculations we just used Equation [3], taking into account all protons in a sphere of 6 Å around the individual amide protons. The use of just Equation 3 is justified by the fact that the HNHA anti-phase $R_2$ relaxation rate is virtually identical to the in-phase HN $R_2$ rate (see the discussion for GB1). We iterated the (unknown) rotational correlation time and found a best

fit for $\tau_c$ =6.3 ns.

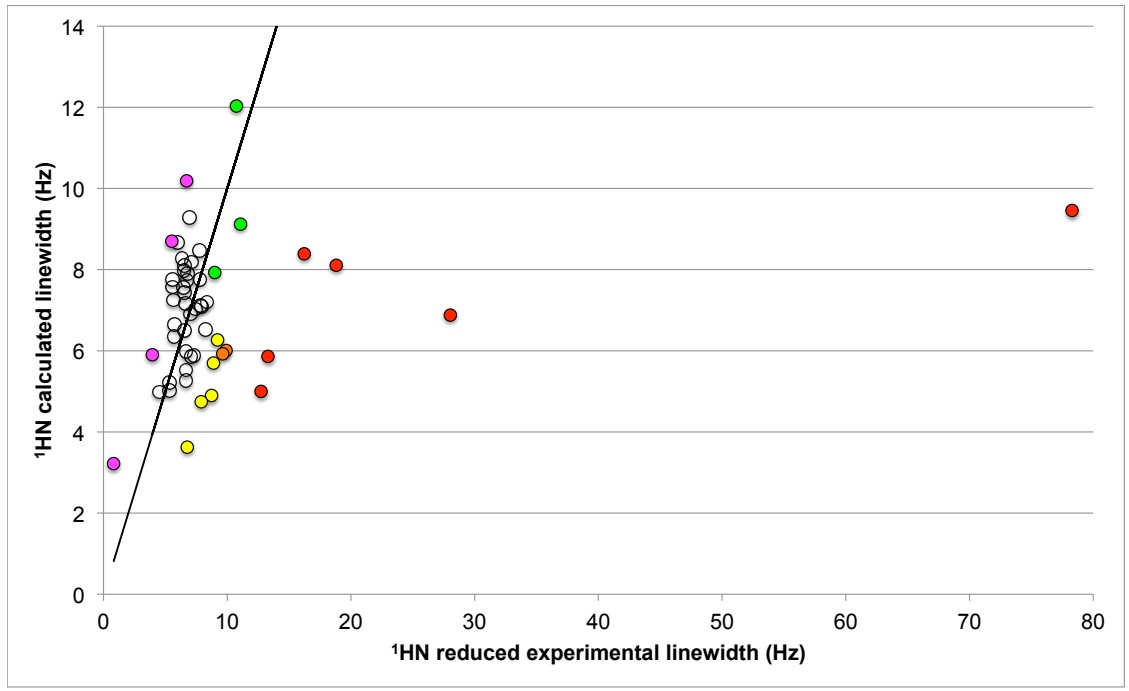

**Figure 6A**. *Calculated [1]HN linewidths for BPTI, using Equation 3, $\tau_c$ =6.3 ns.*

*The experimental linewidth was reduced by the scalar couplings and 2 Hz (see text).*

*The drawn line has a slope of 1. Color coding: Red: D3, T11, C14, K15, G37 and K46. Orange: A16, A40.*

*Yellow:F4, Y10, C38, N44. Green: Y21, G36, G57. Magenta, I19, L29, Y35, R42 and A58.*

**MAGNETIC RESONANCE**
Discussions

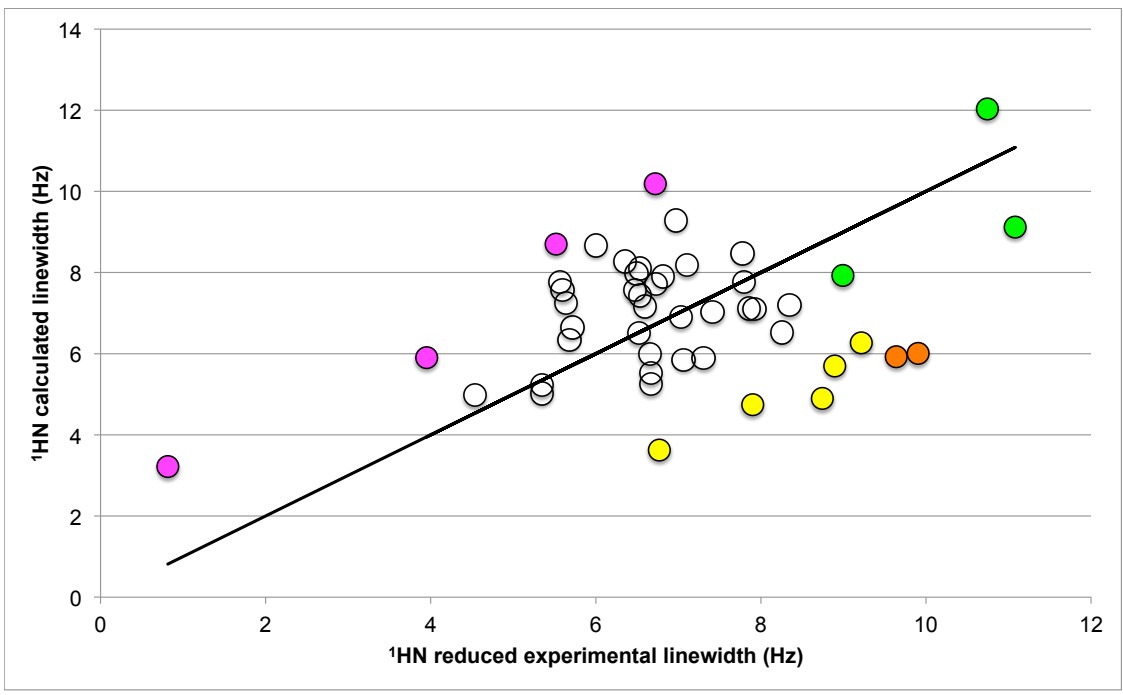

**Figure 6B.** *Enlargement of Figure 6A. Calculated $^1$HN linewidths for BPTI, using Equation 3, $\tau_c$ =6.3 ns. The experimental linewidth was reduced by the scalar couplings and 2 Hz (see text). The drawn line has a slope of 1. Color coding: Orange: A16, A40. Yellow:F4, Y10, C38, N44. Green: Y21, G36, G57. Magenta, I19, L29, Y35, R42 and A58.*


In the $^1$HN linewidth data by itself (i.e. Figure 6A projected on the x-axis), one finds already ample indication of conformational exchange broadening. The red data points for *D3, T11, C14, K15, G37 and K46.* stand out with linewidths upto 80 Hz. Indeed, for red points one does not need a calculation to decide if these resonances are conformationally exchange 370 broadened. In Figure 7, we show the crystal structure of BPTI, where we use the same color coding as in Figure 6.



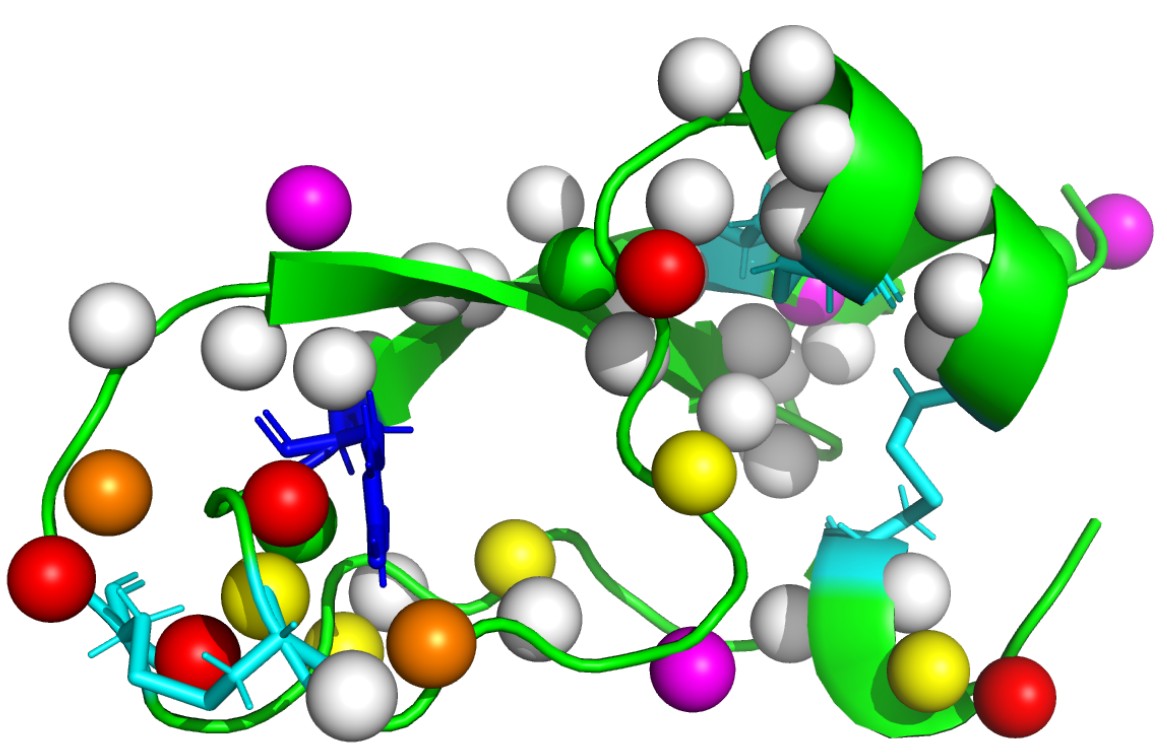

***Figure 7A***. *Crystal structure of BPTI (9pti) with amide protons (spheres) color coded as in Figure 6A. Disulfide bridges are shown in cyan. Tyr35 is shown in blue.*



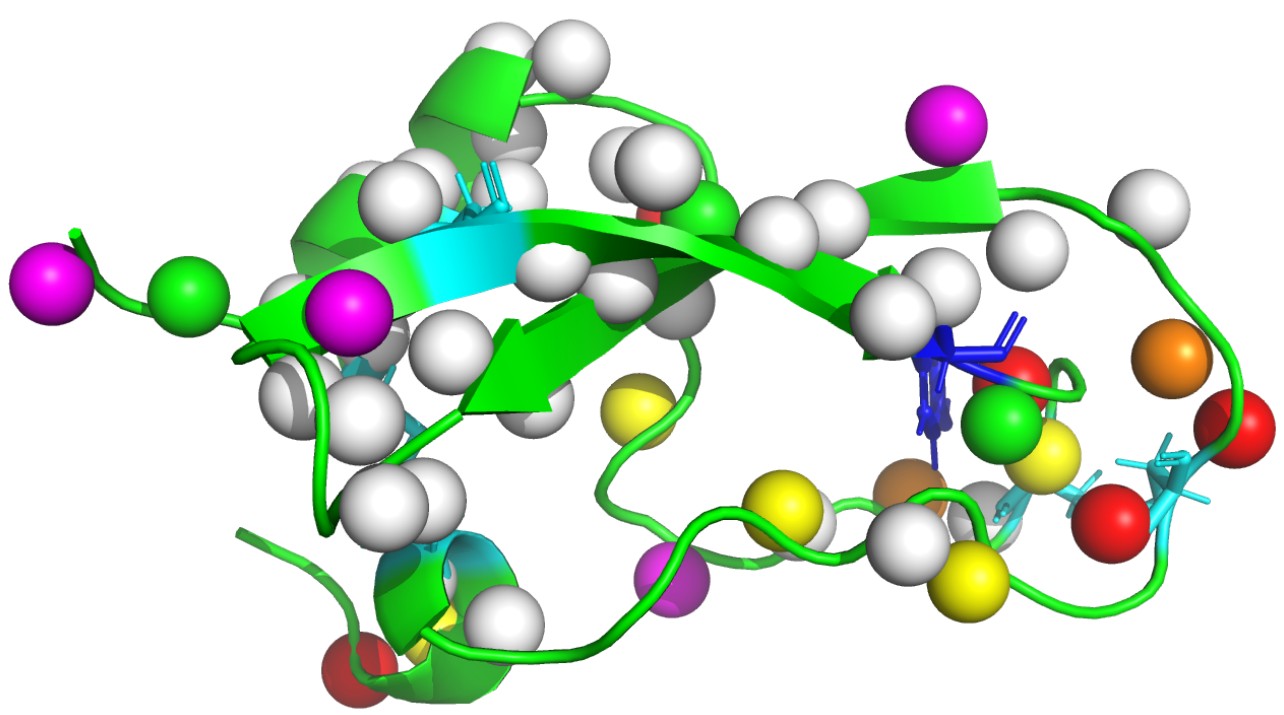

**Figure 7B**. *As Figure 7A, rotated 180 degrees along the vertical axis.*

Three of five excessively broadened resonances belong to region in the left bottom of protein in Figure 7A. This protein area comprises two anti-parallel beta strands with residues 10-15 and 36-40, and harbors the Cys14 – Cys38 disulfide. Returning to Figure 6, one would be hard-pressed to declare the orange points as exchange broadened or not. They are on the edge of the bulk of the distribution. But the calculated data for these points is just 6 Hz; this helps in deciding the matter, suggesting that the experimental data are exchange broadened. The two orange points correspond to A16 and A40, which, significantly, are

in the same area where the red points are in Figure 7A. From the five yellow points in Figure 6 three map in the same area again (10, 12, 38), one is residue Phe4 next to the excessively broad Asp3 (bottom right of Figure 7A). This broadening is likely amine-catalyzed amide proton mass exchange life-time broadening which is so often seen for the N-terminal 3 to 4 residues in proteins. The last yellow point at 6.8 ppm belongs to N44.

The majority of the broadened [1]HN resonances belonging to the red, orange and yellow data points in Figure 6B are

clustered and are not all over the protein. Just by itself, this result suggests that our calculation and its interpretation give rise





to a useable results. But there is more.  In early work, (Szyperski et al., 1993) detected $^{15}$N exchange broadening for residues 14-16 and 38-39 in BPTI. Our current calculations (Figures 6 and 7) point to exactly the same area. (Szyperski et al., 1993) suggest that the $^{15}$N exchange broadening is a result of the Cys14 – Cys38 disulfide isomerization at a stochastic rate of 500 s$^{-1}$ and a superposed conformational process of the entire area with a stochastic rate > 10,000 s$^{-1}$. If we assume that the changes

in chemical shift associated with these conformational changes are the same for $^{15}$NH and $^{1}$HN in terms of ppm, we would expect the 500 s$^{-1}$ process to give rise to slow exchange or resonance doubling in $^{1}$H, which is not observed. Hence it is likely that the $^{1}$HN line widths are sensitive to  the faster process.

The effect of mutations on the $^{15}$N relaxation of BPTI has also been studied. According to (Beeser et al., 1997), fast and slow dynamics is mostly absent in wt-BPTI, with order parameters between 0.8 and 0.9 (except for the C-terminus) and very little

exchange broadening (~ 1Hz), except for two areas (again) 14-15 and 38-40.  (See Figure 4A of (Beeser et al., 1997)). This is in agreement with the data of (Szyperski et al., 1993). The effect of the mutation  Tyr35Gly on $^{15}$N the relaxation parameters was also studied; it exacerbates the broadening in magnitude  (up to ~ 3 Hz) and extent (comprising residues 10-20 and 32-43), (see Figure 4B in (Beeser et al., 1997)). We show Tyr35 in Figure 7.  Interestingly, our $^{1}$H dynamic results also comprise that same *extended* area; it thus seems that the $^{1}$HN resonances are sensitive to extended dynamical processes present in the

wild-type protein, that are only observable by $^{15}$N relaxation after a (predictably) destabilizing mutation.

There are several points that are calculated to be significantly broader than the experiment; these data points are shown in magenta. They comprise I9, L29, R42 and A58. Such outliers would indicate resonances that according to the crystal structure coordinates should be broad (a dense proton environment), but are not broad in the experimental data. That would suggest fast local motion. The $^{15}$N order parameter for Ala58 is small ((Beeser et al., 1997)), which would corroborate the

finding here. However, the $^{15}$N-relaxation data does not show reduced order parameters for I9, L29 and R42. This maybe a genuine difference in $^{15}$N and $^{1}$HN order parameters, or noise. We also have no quick rationale for the experimental broadening for Lys 46 (red, on the top-middle of Figure 7A) and the calculated rate for Asn 44 (yellow, just below it). Previous work $^{15}$N relaxation work does not show anything particular for these residues. But, the resonance of K46 is actually missing at the contour level of the spectrum in Figure 5; so there is no question that something is going on there.  We may speculate that

small motions of the ring of Phe45, which hovers above amides 44 and 46, can translate in changing ring-current shifts, causing broadening for these resonances, which is not due to an actual spatial change at the level of the amides themselves. A ring-current-driven mechanism could be consistent with the lack of broadening effects in the $^{15}$N spectral data: ring current effects are, as expressed in Hz, 10 times larger for $^{1}$H than for $^{15}$N . Hence, varying ring current shifts are apt to cause much more "conformational" exchange broadening for $^{1}$HN than for  $^{15}$NH. Last, there are three points Y21, G36 and G57, colored green.

The calculations identify them as broad as well, suggesting that the broad linewidth is intrinsic.  According to the $^{15}$N data, no broadening is occurring for those residues either. It is significant for assessing the value of our calculations,  that the two right-most green points in Figure 6B (G36 and G57) would be identified as exchange broadened from the experimental $^{1}$HN linewidth distribution, but that the calculations indicate that they are not.



But how do we do with our calculations *within* the bulk of the distribution? Figure 6B shows that it is not good at all
and not much worse than for GB1 (Figure 4B). There is hardly a correlation between experiment and calculation – the
calculated values all lie around 7 Hz, while the experimental values vary almost a factor of two. At the moment we have no
explanation for this, merely suggesting that there is much room for improvement, likely in measuring / calculating the [1]HN-
[1]HX order parameters.  We note that we used a BPTI crystal structure with 1.2 Å resolution,  which has a coordinate precision
of ~ 0.2 Å (DePristo et al., 2004). This can give rise to considerable errors in $R_2$ calculations. For example, a HN(i) to HA(i-
1) distance of nominally 2.2 Å in a beta sheet structure (Wüthrich, 1986) may be incorrect by 9%, and produce a 68% error
for the $R_2$ relaxation contribution. There is work to be done here, for sure. Nevertheless, just considering the *outliers* of the
distribution appears to result in relevant dynamical information.




## 4. Conclusion

We developed computer programs to predict amide proton line widths from (crystal) structures. We calibrate our programs by comparing computational and experimental results for GB1, using $^{15}$N-$^1$H HSQC and semi-selective T$_{1rho}$ experiments. We find that we can predict the correct range of $^1$HN R$_2$ relaxation rates from a crystal structure using a Karplus equation and a program based on just one relaxation equation. We deduce that GB1 has fairly low average $^1$HN order parameters (0.8), in broad agreement with what was found by others from $^{15}$N relaxation experiments. We apply the program to the BPTI crystal structure and compare the results with a $^{15}$N-$^1$H HSQC spectrum of BPTI. After adjusting the correlation time, we find from the outliers in the distribution a cluster of conformationally broadened $^1$HN resonances that belong to an area for which broadened $^{15}$NH resonances have been previously reported. Thus, our approach can yield important dynamical data. We feel that this approach may be useful to glean insights into the dynamical properties of larger biomolecules for which high-quality $^{15}$N relaxation data cannot be recorded. The semi-selective T$_{1rho}$ experiments are also not difficult to perform and are also suitable for application to larger molecules. Comparing these relaxation data (T$_{1rho}$ and the $^1$HN linewidth) for proteins in different states or complexation forms, is likely much more interesting. Perhaps the dynamical differences can be tied to functional properties, as has been carried before for small proteins, but much less so for the larger ones. The theory of $^1$HN R$_2$ for proteins is not iron-clad; issues such as "like" and "unlike" R$_2$, "in-phase/antiphase" relaxation, "selective" and "unselective" R$_1$ rates and cross-correlated R$_2$ relaxation all play roles in these issues. As a by-product of our "calibration" work for GB1, we help resolve most of those (sometimes contentious) issues.

## 5. Acknowledgements

Dr. Gottfried Otting has graciously provided me with remote instrument time at the Canberra NMR facility, and the use of his sample of $^{13}$C,$^{15}$N labeled GB1. I thank Professor Arno Kentgens (Nijmegen) for helpful discussions on solid state NMR powder patterns. I have taken advice from Professors Raphael Bruschweiler (Ohio State), Gareth Morris (Manchester) and Drs. Ad Bax (Bethesda), Bernhard Brutscher (Grenoble) and Anaya Majumdar (Baltimore).

## 6. Code/Data availability.

The Fortran90 computer codes are available from the author and will be deposited at https:://github.com.

The (Bruker) data directories for the GB1 experiments will be deposited at the Biological Magnetic Resonance Bank.

## 7. Author contributions.

ERPZ conceived and wrote the paper. He wrote all computer codes and remotely carried out the GB1 NMR experiments.

## 8. Conflicting interests

None.




### 9. Supplemental materials

Tables with Sparky linewidth/integration lists for GB1 and BPTI, as well as data fits for the semi selective $T_{1rho}$ experiments, as well the pulse sequence for that experiment, are given in the Supplemental Materials.





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



## 11. Appendix.

### Computer programs based on Eqs. 3 ($R_2$) and 6 ($R_1$).

The program requests a PBD file for which protons are available. The program makes an internal copy of the pdb file. It requests the radius of the sphere of protons to consider, the rotational correlation time, and the spectrometer. The code consists

of two loops; the outer loop advances over the amide protons one by one. The inner loop scans the copy of the coordinates and finds all protons (including HN) around the HN at hand for the radius defined. It co-adds all $R_2$ rates according to Equation 3 and 12.

The program calculating the $^1$HA $R_1$ and $^1$HN $R_1$ rates are almost identical to the $R_2$ program, with changed equations.

The linear combinations of $R_2$(HN) and $R_1$(HA) , as governed by Equation 6, was carried out in a spreadsheet, taking into

account different amounts of in-phase / anti-phase admixtures based on the integration of model computations with different scalar couplings and acquisition times.

All programs are written in Fortran90 , and contain no references to outside libraries. The source codes are available from the author and will be deposited at https://github.com.

**Computer program based on Eqs. 12 and 13 (below).**

Proton-proton cross-correlated $R_2$ relaxation between just two dipolar vectors **ij** and **ik** is, adapted from (Goldman, 1984) and (Fischer et al., 1998)

$$R_{2CC}^{ij-ik} = \frac{1}{10}\left(\frac{\mu_0}{4\pi}\frac{\gamma_i\gamma_j\hbar}{r_{ij}^3}\right) \times \left(\frac{\mu_0}{4\pi}\frac{\gamma_i\gamma_k\hbar}{r_{ik}^3}\right)$$

$$\times P_2\left(\cos\theta_{ij-ik}\right)\left\{5\tau_c + \frac{9\tau_c}{1+\omega_H^2\tau_c^2}\right\}$$

[13]

where $\theta_{ij-ik}$ is the angle between the two vectors **ij** and **ik**.

The total **$R_2$** relaxation for proton **i** is then given by

$$R_2^{i-total} = R_2^{ij} + R_2^{ik} \pm R_{2CC}^{ij-ik}$$

[14]

However, these individual line widths can only be observed if the transitions for the **$H_i$** multiplet are resolved by J-coupling (and/or residual static dipolar coupling). For amide protons, this will not be the case, and one expects an inhomogeneous line consisting of the superposition of many narrow and broader Lorentzian lines corresponding to a multi-spin expansion of Eq [14].





To my knowledge, there is no closed equation describing $R_2$ cross-correlated relaxation for more than two dipolar vectors.
To arrive at an estimation for the effects in a multi- proton spin system, we start from a "solid state NMR" point of view. We calculate $B_{loc(i)}^{\Omega}$, the net local magnetic field at center proton $\textbf{\textit{i}}$ due to the surrounding protons $\textbf{\textit{j}}$ (Slichter, 1992) in certain orientation of the magnetic field with respect of the molecule:

$$B_{loc(i)}^{\Omega} = \frac{\mu_0 \hbar}{4\pi} \sum_{j \neq i}^{j=M} \Phi_{\pm} \left( \frac{\gamma_i \gamma_j}{r_{ij}^3} \right) P_2 \left( \cos \theta_{ij} \right) \qquad [15]$$

Here, $\theta_{ij}$ is the angle between the internuclear vector $\textbf{\textit{ij}}$ and the magnetic field direction $\Omega$ in the molecular frame. $\Phi_{\pm}$
represents a certain configuration of the signs of the surrounding dipoles $\textbf{\textit{j}}$. For instance, for 10 protons one has 1024 different configurations. If one varies the magnetic field direction according to a sphere distributions and adds the results one obtains the cross-correlated powder pattern for that value of $\Phi_{\pm}$. Subsequently one co-adds all powder patterns for different values of $\Phi_{\pm}$, and normalizes, to arrive at the "cross correlated" dipolar powder pattern for the [1]HN under consideration.

It is the time-dependence of $\textbf{\textit{B}}_{loc}$ as caused by molecular motion that drives the solution NMR dipolar relaxation. The $R_2$
relaxation is then obtained as the second moment of the (cross-correlated) powder pattern (Slichter, 1992):

$$R_2^{solution} = 4\tau_C \sum_{\Omega} \left( \left\langle B_{loc} \right\rangle - B_{loc}^{\Omega} \right)^2 \qquad [16]$$

where the brackets indicate average.

610        The computer program requires as input a "protonated" PDB file (HN for amides), the radius of the sphere of protons around the amide protons, the rotational correlation time and the spectrometer frequency. Basically, the program consists of four nested loops: amides, protons around amides, permutation of dipole signs of these surrounding protons, and rotation of the magnetic field vector in the molecular frame.

A set of 10 nested loops permutes the dipolar signs of the closest 10 hydrogens (1024 distributions). The more remote
hydrogens in the sphere (if any) have their dipolar signs assigned according to a 50% random chance.

At the inner most level, a closed loop generates an isotropic spherical distribution (5000 orientations) for the (unit) "magnetic field" vector (http://corysimon.github.io/articles/uniformdistn-on-sphere/).

The angle between the (unit) magnetic field vector and the dipolar vector between HN and the surrounding proton is computed as the arccosine of the (normalized) dot product of those vectors.
The local dipolar field of an individual surrounding proton at [1]HN is then calculated according to Equation 14.

This is repeated for all surrounding protons in the shell and co-added.



At this stage the program has the local field for a certain $^1$HN, in a certain orientation, for a certain permutation of surrounding dipole signs. This repeated for all 5000 orientations, so that it obtains the $^1$HN powder pattern for a certain permutation of the surrounding dipole signs.

Subsequently the corresponding solution NMR line width is computed from this distribution by the method of second moments (Equation 15).

The next step is to repeat this for all 1024 permutations. The line widths are all added and normalized yielding the inhomogeneous linewidth. The inverse line widths, which are proportional to the peak height, are also added.

After that the outer loop advances to the next HN.

The program is written in Fortran90 , and contains no references to outside libraries. The source code is available from the author and will be deposited at https://github.com.



*Comparison of cross-correlated and non-cross correlated R₂ relaxation.*

Figure A1 shows a comparison between the line widths computed for GB1with and without cross correlation. The data with

cross correlations was computed using the "solid state" approach above, taking into account all protons within a sphere of 6

Å.    For the non-cross correlated data, we used the same approach, but instead of calculating 1024 different specific

permutations, we used 1024 random distributions of surrounding dipoles, and averaged those. The Figure shows that there can

be upto 1.5 Hz differences between the two methods, but there is no systematic change, and is of no relevance to our current

level of computational precision.  But if future calculations ask for refinement, one must include the cross correlations.

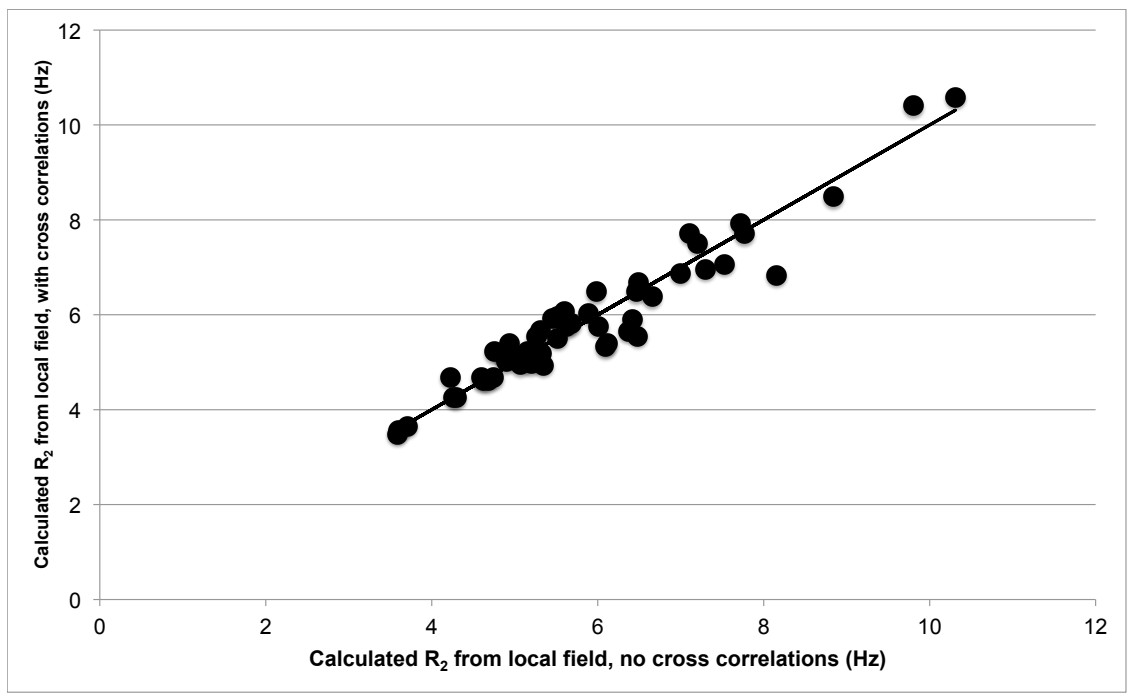

*Figure A1.*

*The effect of $R_2$ dipolar cross correlations on the $^1HN$ $R_2$  for GB1.*



### 2. Rotational correlations times


(Daragan and Mayo, 1997) noted a deviation between the $\tau_C$ values calculated for proteins from the Stokes-Einstein relation and the experimental values, which were then known for proteins smaller than 18 kDa. Fitting to that data, they obtained the empirical $M_r$ vs. $\tau_C$ relationship:

$$\tau_C = N_{RES}^{0.93} \frac{9.18 \times 10^{-3}}{T} \exp\left(\frac{2416}{T}\right) \quad (ns) .$$
[17]

with T in $^0$K. We collected several more experimental rotational correlation times (see Table A1), and find that equation 17 also holds outside the range for which it was developed.

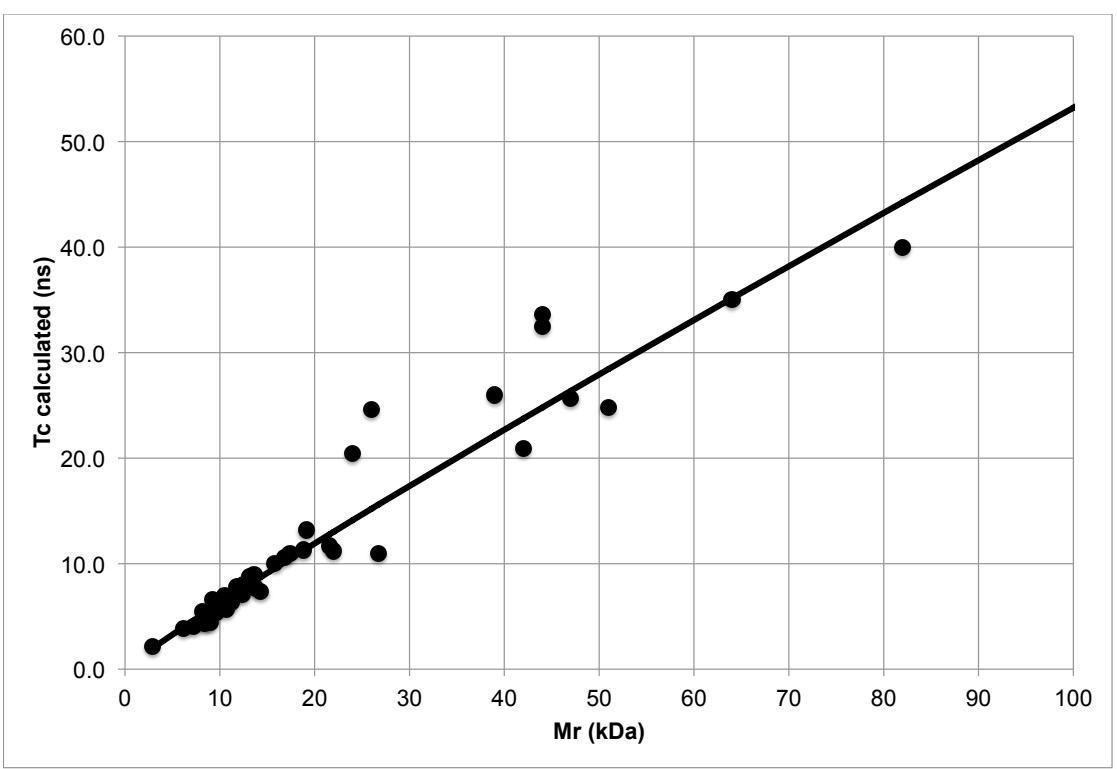

**Figure A2**. *Experimentally determined rotational correlation times for different protein masses (dots) as collected from the literature, measured or corrected to 298K using the known temperature dependency of the water viscosity (see Table A1). The drawn line was computed using Equation 17, also at 298K.*





| Protein | Mr | $\tau_c$ | Temperature | $\tau_c$ @ 25 $^0$C in $H_2O$[a] |
|---|---|---|---|---|
| | (kDa) | (ns) | ($^0$C) | (ns) |
| Xfin-zinc[b] | 2.93 | 2.4 | 20 | 2.1 |
| BPTI[b] | 6.16 | 4.4 | 20 | 3.9 |
| B-domain[c] | 7.2 | 4.05 | 25 | 4.0 |
| Eglin[b] | 8.15 | 6.2 | 20 | 5.5 |
| Calbindin-D9k[b] | 8.43 | 4.9 | 20 | 4.3 |
| Calbindin-D9k[b] | 8.43 | 5.1 | 20 | 4.55 |
| Ubiquitin[b] | 8.54 | 5.4 | 20 | 4.8 |
| Ubiquitin[c] | 9.0 | 4.4 | 25 | 4.4 |
| StR82[c] | 9.2 | 6.6 | 25 | 6.6 |
| Cytochrome b5[b] | 9.61 | 6.1 | 20 | 5.4 |
| Barstar[b] | 10.14 | 7.4 | 20 | 6.5 |
| TR80[c] | 10.5 | 7.0 | 25 | 7.0 |
| HR2873B[c] | 10.7 | 5.7 | 25 | 5.7 |
| DvR115G[c] | 10.9 | 6.5 | 25 | 6.5 |
| VfR117[c] | 11.2 | 6.3 | 25 | 6.3 |
| MrR110B[c] | 11.8 | 7.8 | 25 | 7.8 |
| BcR147A[c] | 11.9 | 7.2 | 25 | 7.2 |
| SyR11[c] | 12.4 | 7.1 | 25 | 7.1 |
| VpR247[c] | 12.5 | 8.1 | 25 | 8.1 |
| BcR97A [c] | 13.1 | 8.8 | 25 | 8.8 |
| PfR193A[c] | 13.6 | 9.0 | 25 | 9.0 |
| SoR190[c] | 13.8 | 7.7 | 25 | 7.7 |
| Lysozyme[b] | 14.32 | 8.3 | 20 | 7.3 |
| ER541-37-162[c] | 15.8 | 10.0 | 25 | 10.0 |
| NsR431C[c] | 16.8 | 10.6 | 25 | 10.6 |
| Interleukin-1b[b] | 17.4 | 12.4 | 20 | 11.0 |
| ER540[c] | 18.8 | 11.3 | 25 | 11.3 |
| Leuk Inh Fact[b] | 19.1 | 14.9 | 20 | 13.2 |

Table A1. Experimental $M_r$ and $\tau_c$ data.



| | | | | |
|---|---|---|---|---|
| HIV-1[b] | 21.58 | 13.2 | 20 | 11.7 |
| HIV protease[d] | 22 | 10.7 | 27 | 11.2 |
| Trp-repressor[b] | 24 | 23.1 | 20 | 20.4 |
| DnaK SBD[e] | 26 | 22.0 | 30 | 24.6 |
| Savinase[b] | 26.7 | 12.4 | 20 | 11.0 |
| lS- dehydrase[f] | 39 | 18.4 | 42 | 26.0 |
| Maltose BP[g] | 42 | 16.2 | 37 | 20.9 |
| Hsc70NBD[h] | 44 | 30.0 | 30 | 33.6 |
| DnaK NBD[i] | 44 | 29.0 | 30 | 32.5 |
| CAP[j] | 47 | 22.0 | 32 | 25.7 |
| PMM/PGM[k] | 51 | 20.0 | 35 | 24.8 |
| Hemoglobin[l] | 64 | 32.0 | 29 | 35.0 |
| Malate Synthase[m] | 82 | 37.0 | 37 in D2O | 40 |


*Legend to Table A1:*

(a) converted using (Weast, 1973)

(b) Values listed in  (de la Torre et al., 2000)

(c) Experimental values from the North East Structural Genomics initiative  listed at

http://www.nmr2.buffalo.edu/nesg.wiki/NMR_determined_Rotational_correlation_time

(d) A. Bax, personal communication

(e) Experimental value from (Bertelsen et al., 2009)

(f)  Experimental value from  (Copié et al., 1996)

(g) Experimental value from (Gardner et al., 1998)

(h) Experimental value from Weaver, D., Ph.D. thesis, University of Michigan, 2010

http://hdl.handle.net/2027.42/75930

(i) Experimental value from (Korzhnev et al., 2004)

(j) C. Kalodimos, personal communication

(k) Experimental value from (Sarma et al., 2012)

(l) Experimental value from (Song et al., 2007)

(m) L. Kay and V. Tugarinov, personal communication.