# Peer review of "Insights into Protein Dynamics from 15N-1H HSQC"

_Magnetic Resonance, 2021_

## Referee Comment (RC1)

This paper has an ambitious objective: avoid all the niceties of measuring various $^{15}N$ relaxation rates, on the grounds that these sophisticated experiments do not offer sufficient sensitivity for large biomolecules in low concentrations, and use only the proton linewidths in HSQC spectra, supplemented by semi-selective $T_{1rho}$ measurements.

The author is far from naïve and discusses most hurdles in lucid detail. "Amide proton linewidths may be affected by a plethora of mechanisms, which we will try to unravel in this work." The crucial questions are indeed: "is a narrow line narrow because it has few dipolar neighbors, or is it motionally narrowed? Is a broad line broad because of conformational exchange, or because it has a dense proton environment?"

Unfortunately, the outcome of the analysis is disappointing. The problems are summarized in the conclusions: "The theory of 1HN R2 for proteins is not iron-clad; issues such as "like" and "unlike" R2, "in-phase/antiphase" relaxation, "selective" and "unselective" R1 rates and cross-correlated R2 relaxation all play roles in these issues."

I cannot agree more. The main problem is that these issues cannot be resolved by establishing clear boundaries. Thus, one cannot easily choose between "like" and "unlike" R2, since the question if two chemical shifts can be considered to be degenerate depends on the linewidths. There is of course a grey zone between "like" and "unlike"; indeed, an equation describing a smooth transition between identical and non-identical spins has been given (Goldman, 1988). The relative weights of "in-phase" and "antiphase" contributions to transverse relaxation depend on the scalar couplings, the lifetimes of the signals, and on truncation of the signals if the observation ("acquisition time") has a limited length. The distinction between "selective" and "unselective" R1 rates of neighboring scalar-coupled protons (that contribute to transverse relaxation of antiphase terms) depends on the degree of saturation, the breadth of the rf irradiation (the statement "hence the 5 kHz r.f. field "hits" those HA whereas the 500 Hz r.f. field does not" does not leave any room for a grey area.) The spin temperature of the surrounding bath ("hot" if saturated, or "cold" if in thermal equilibrium) is not uniform since there must be an offset-dependent grey zone between "hot" and "cold". Methods designed to return the water magnetization to +z during the FID are never perfect. Contributions of a manifold of neighboring protons (as many as "40 protons in a 6 Å sphere around an amide proton"!) are certainly not additive, but it is tricky to extend consideration of cross-correlated fluctuations to a manifold of densely packed neighboring protons.

To our relief, the author frankly admits that he has not solved all problems: "There is hardly a correlation between experiment and calculation – the calculated values all lie around 7 Hz, while the experimental values vary almost a factor of two. At the moment we have no explanation for this …"

Statements like "Experimental data points larger than computed ones are harder to explain" amount to admitting a failure of the analysis.

It is therefore surprising to read "With this, we have arrived at our goal: we show that we *can* extract motional information from the 1HN linewidths in a HSQC spectrum by making simple calculations based on a crystal structure."

To my regret, I cannot recommend acceptance of this paper in "Magnetic Resonance", nor in any other journal. It seems useful however that it will remain accessible on "Magnetic Resonance Discussions", along with this critical review, since "Interactive comments are posted alongside the preprint and will remain permanently archived, publicly accessible, and fully citable."

A few details:

The scalar couplings, whether resolved or not, should not be "subtracted" but must be properly de-convoluted.

The assumption that anisotropy of rotational diffusion can be neglected because "relaxation vectors" point in many directions seems a bit superficial.

I would prefer the use of indices like in $^{15}N^H$ and $^1H^N$ or, since isotopes are obvious, simply $N^H$ and $H^N$ rather than the ambiguous notation NH and HN

Since indices on indices are hard to print, why not use $^3J(H^NH^A)$ rather than $^3J_{HNHA}$ ?

Likewise, it would be better to use symbols such as "$R_1(H^A)$ contributing to $R_2(H^N)$" rather than long-winded phrases like "R1 relaxation rate for 1HA contributing to the effective R2 relaxation rate of 1HN."

It would be better to speak about "internuclear vectors" instead of "relaxation vectors"

There are a few minor spelling errors like: offfeset, KHz for kHz

What is meant by "Nevertheless, the protein is not perdeuterated"?

What is meant by "three of five *excessively* broadened resonances"?

Do SeM and Sem both mean seleno-methionine?

---

## Referee Comment (RC2)

The author has made a commendable effort to accurately calculate ¹H line widths in proteins, using GB1 and BPTI as test systems. As noted by the author, many theoretical issues are difficult to clearly resolve in doing the calculations. The author ultimately concludes about BPTI: "But how do we do with our calculations within the bulk of the distribution? Figure 6B shows that it is not good at all and not much worse than for GB1 (Figure 4B). There is hardly a correlation between experiment and calculation". Clearly, if this was the end of the story, then the paper hardly could be publishable.

I am less pessimistic than the author however. Given current technology, the goal of calculating ¹H line widths quantitatively probably is unattainable. Even if all the challenges presented by relaxation theory were solved, structural differences in solution compared to the crystal structure, even if minor, have a large effect on relaxation, owing to the $1/r^6$ dependence of the dipole coupling (as noted by the author). To obtain quantitative agreement, averaging over an MD simulation (or some other ensemble) would be needed. But if we had such a reliable simulation or ensemble, we would hardly need to look at the line widths to discern dynamics.

So what can be done? The distributions of the calculated and experimental line widths for BPTI shown in Figure 6B are similar (ignoring the clearly exchange broadened residues in Figure 6A) Thus, given the effect of structural noise, etc. I think it is reasonable to regard the different residues then as representing random samples of different magnetic environments (that is, the calculated linewidth for residue X is not the experimental linewidth for residue X, but the experimental linewidth for a residue Y that happens to have a ¹H density in solution similar to residue X in the crystal). The similar distributions then between the calculated and experimental measurements therefore would suggest that the author has gotten a lot of the physics right, but the correct physics is disguised by structural/dynamical noise. The standard deviation for the calculated linewidths is ~1.6 Hz (I estimated values from the figure). I have taken the liberty of rescaling Figure 6B and drawing the 1 sigma and 2 sigma contours (in practice, it might make sense to winsor the data a bit, but that is another topic). The similarity of the distributions is evident. In this scenario, the best that can be done is to identify outliers from the distribution, which the author has done. Thus, the author's work strikes me as a reasonable achievement.